# CIKQA: Learning Commonsense Inference with a Unified Knowledge-in-the-loop QA Paradigm

**Hongming Zhang[1,2], Yintong Huo[3], Yanai Elazar[4,5], Yangqiu Song[1], Yoav Goldberg[4,5], Dan Roth[2]**

[1]HKUST, [2]UPenn, [3]CUHK, [4]Bar Ilan University, [5]AI2

{hzhangal,yqsong}@cse.ust.hk, ythuo@cse.cuhk.edu.hk
{yanaiela,yoav.goldberg}@gmail.com, danroth@seas.upenn.edu

## Abstract

Recently, the community has achieved substantial progress on many commonsense reasoning benchmarks. However, it is still unclear what was learned from the training process: the knowledge, how to do inference, or both? We argue that due to the large scale of commonsense knowledge, it is infeasible to annotate a large enough training set for each task to cover all commonsense for learning. Thus we should separate the commonsense knowledge acquisition and inference over commonsense knowledge as two separate tasks. In this work, we focus on investigating models' commonsense inference capabilities from two perspectives: (1) Whether models can know if the knowledge they have is enough to solve the task; (2) Whether models can learn commonsense inference capabilities, that generalize across commonsense tasks. We first align commonsense tasks with relevant knowledge from commonsense knowledge bases and ask humans to annotate whether the knowledge is enough or not. Then, we convert different commonsense tasks into a unified question answering format to evaluate models' generalization capabilities. We name the benchmark as Commonsense Inference with knowledge-in-the-loop Question Answering (**CIKQA**).

## 1 Introduction

Understanding human language requires both the language knowledge (e.g., grammar and semantics) and world knowledge, which can be further divided into factual and commonsense knowledge (Katz and Fodor, 1963). Recently, the community has made great progress on helping machines acquire and apply language and factual knowledge. However, how to help machines acquire and inference over commonsense is still unclear. To answer this question, many commonsense reasoning datasets (Roemmele et al., 2011; Sakaguchi et al., 2019; Talmor et al., 2019; Zellers et al., 2019; Lin et al., 2020) have been proposed. Even though

they target different knowledge types, modalities, and come in different formats, they often follow a standard supervised learning setting, and aim at helping machines to solve a specific task with the training data. However, two limitations of this learning paradigm have limited the development of commonsense reasoning systems.

First, there is no clear separation between knowledge and inference. As discussed in (Elazar et al., 2021), a common phenomenon is that larger training data will lead to better performance, mainly because richer knowledge is covered. However, due to the large scale of commonsense knowledge, it is infeasible to annotate a large enough training data for each task, and the responsibility of the training data should be teaching models how to do inference rather than acquiring the commonsense knowledge. Several recent works have explored using structured knowledge for commonsense reasoning tasks (Lin et al., 2019; Lv et al., 2020; Paul and Frank, 2020). However, as these works did not clearly analyze the coverage of the structured knowledge (i.e., knowledge graphs), it is still unclear what the performance means, better knowledge coverage or better inference capability. To dig into what is behind this learning process, we propose to equip each question with supporting knowledge. By doing so, we could evaluate whether models can know if there provided knowledge is sufficient or not and how well they can conduct inference over the provided knowledge to solve the task.

Second, the supervised learning may force the model to learn the distribution of the training data rather than a universal inference model. As a result, the model may perform well on the test set that follows the same distribution but fail on other tasks (Kejriwal and Shen, 2020). Previously, as different tasks have different formats, it is hard to evaluate the generalization ability of commonsense reasoning models. Motivated by existing

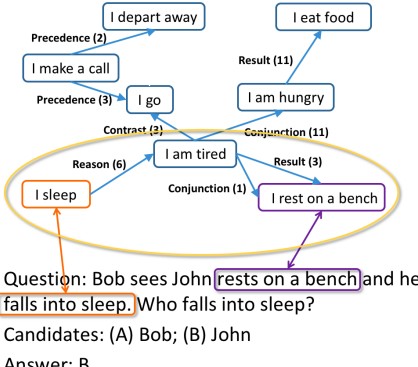

Question: Bob sees John rests on a bench and he falls into sleep. Who falls into sleep?
Candidates: (A) Bob; (B) John
Answer: B

Figure 1: **CIKQA** demonstration. Models need to learn that all pronouns "I" refer to the same person and then solve the question based on the knowledge that "one may fall into sleep if he/she rests on a bench."

trends of using a unified format (i.e., question answering) for different tasks (Khashabi et al., 2020), we propose to convert various commonsense reasoning tasks into a unified QA format such that we can easily and fairly evaluate the generalization ability of learned commonsense reasoning models.

Combining these two lines of effort, we propose a new commonsense inference evaluation benchmark Knowledge-based Commonsense Inference with QA (**CIKQA**). An example is shown in Figure 1. We equip several popular commonsense reasoning tasks with the supporting knowledge from existing commonsense knowledge graphs and convert them into a unified QA format. We leverage human annotation to label whether the provided knowledge is enough to answer the question. With **CIKQA**, we are interested in answering three questions: (1) Whether current models can distinguish the knowledge is gold or not; (2) Whether current models can learn to conduct inference over provided knowledge; (3) Can current commonsense inference models generalize across different commonsense reasoning tasks.

Experiments with several recent knowledge-based commonsense reasoning models and a proposed baseline **JointI**, which jointly encodes the knowledge and question with a single model, show that even though inference over commonsense knowledge is challenging, models can learn to conduct simple inference after training with a few examples and better answer the questions than not using the knowledge. As a comparison, learning to distinguish gold knowledge is still a more challenging task. Last but not least, even though current models demonstrate the encouraging generalization ability across three relatively simple

tasks, they still cannot learn complex inference (i.e., compare multiple paths) very well. We hope that our benchmark could motivate more advanced commonsense inference methods in the future.

## 2 Related works

To help machines understand commonsense, the community has devoted great efforts to constructing commonsense knowledge bases with either crowdsourcing (e.g., ConceptNet (Liu and Singh, 2004) and ATOMIC (Sap et al., 2019)) or information extraction techniques (e.g., ASER (Zhang et al., 2020)). Typically, crowd-sourced knowledge bases have higher quality, but the auto-constructed ones have larger coverage. Besides acquiring the commonsense knowledge, the community also developed many commonsense reasoning datasets to test models' commonsense reasoning abilities. Even though these datasets may have different *formats* (e.g., slot fitting in Winogrande (Sakaguchi et al., 2019) and question answering in CommonsenseQA (Talmor et al., 2019)), *knowledge types* (e.g., causal commonsense in COPA (Roemmele et al., 2011) and numerical commonsense in NumerSense (Lin et al., 2020)), or *modalities* (e.g, visual commonsense in VCR (Zellers et al., 2019) and textual commonsense in many others), they all follow a standard supervised learning setting, and aim at helping machines to solve a specific commonsense task in an end-to-end manner. Given this setting, it is often difficult to tell what has been learned during the training process. Was it used to acquire commonsense knowledge, learn to conduct commonsense inference, or both? Such ambiguity limits our progress in solving these commonsense reasoning tasks. In this work, we connect the efforts on commonsense acquisition and inference by creating a commonsense inference benchmark **CIKQA** , where the models can focus on learning to do the inference over the supporting commonsense knowledge graph (KG).

Answering questions in natural language based on a knowledge base (KB) has been a mature research topic in the NLP community, which is also known as the KBQA problem (Clark et al., 1999; Yih et al., 2015, 2016; Usbeck et al., 2017; Cui et al., 2017). Previous works mainly focus on factual knowledge, which is stored in the format of triplets, and the main challenge is how to parse the question and then precisely and effec-

tively identify the correct path over a large-scale KB to do the inference. Compared with inference over factual knowledge, inference over commonsense knowledge brings the following unique challenges: (1) Commonsense is a kind of preference rather than fixed knowledge, which typically involves the comparison of several candidates. As a result, the ideal commonsense reasoning process could involve the comparison of multiple paths; (2) Commonsense is about daily events or objects rather than named entities, and thus it is difficult to find an exact node from the commonsense KB that matches the question and we may need to conduct inference based on the partial match (i.e., the extracted nodes are relevant but not identical).

## 3 Task Formulation

In **CIKQA**, to encourage a generalizable commonsense inference model, we equip each question with a supporting knowledge graph $G$. and follow previous works (Khashabi et al., 2020; Cohen et al., 2020; Wu et al., 2020; Du and Cardie, 2020) to unify all selected tasks as a binary question answering problem ($Q$, $A_1$, $A_2$). As the auto-extracted knowledge could contain noise or may not cover all the essential knowledge for answering the question and humans are capable of saying "I do not know" when they do not know how to answer a question, we leverage human annotators to annotate whether the knowledge is gold (i.e., accurate and enough) for answering the question and test whether current models have the same commonsense reasoning capability of distinguishing the gold knowledge as humans. Details about task selection, format unification, support knowledge extraction, and annotation are as follows.

### 3.1 Task Selection and format Unification

In **CIKQA**, we select the following four popular commonsense reasoning tasks:

1. HardPCR: The hard pronoun coreference resolution (HardPCR) task is one of the most famous commonsense reasoning tasks. For each question, a target pronoun and two candidate mentions are provided, and the task is to select the correct mention that the pronoun refers to. Careful expert annotations are conducted to get rid of the influence of all simple linguistic rules and ask the models to solve the problem with commonsense reasoning. In **CIKQA**, we include instances from WSC (Levesque

et al., 2012), DPR (Rahman and Ng, 2012), and WinoGrande (Sakaguchi et al., 2020). To create a question regarding the target pronoun, we first find the sentence that contains the target pronoun and then determine whether the pronoun refers to a person or an object. If it is a person, we will ask who participates. Otherwise, we will ask what participates.

2. CommonsenseQA (Talmor et al., 2019): CommonsenseQA is a commonsense question answering dataset. For each question-answer pair, four relevant but wrong concepts are used as the other candidates, and the models are required to select the correct one out of five candidates. In **CIKQA**, we randomly sample a negative answer to make it a binary choice task, which is consistent with other datasets.

3. COPA (Roemmele et al., 2011): COPA focuses on evaluating whether models can understand the causality between events or not. For each head event, two candidate tail events are provided, and models are asked to predict the one caused by or the reason for the head event.

4. ATOMIC (Sap et al., 2019): The last one is the commonsense knowledge base completion. Given a head concept (e.g., "eat food") and a relation (e.g., "cause"), we want to predict the tail concept. In **CIKQA**, we focus on predicting edges of ATOMIC.

For COPA and ATOMIC, as they are essentially predicting the relations between two events or states (e.g., "PersonX eats"-*Causes*-"PersonX is full"), for each edge, we randomly sample another event or state (e.g., "PersonX is hungry") as the negative tail and ask the model to select the correct one. To make the task challenging and avoid sampling irrelevant events or states, we require the sampled negative event or state to be connected with the head event or state with a different edge. For each type of relation, we write a simple pattern to generate the question. For example, for the "Causes" relation, we will ask "What can be caused by 'PersonX is hungry'?" Demonstrations of instances in original datasets and their transformed questions and candidate answers are presented in Appendix Section C.

### 3.2 Supporting Knowledge Extraction

As mentioned in Section 1, a limitation of existing commonsense reasoning benchmarks is that there

is no clear boundary between knowledge and inference such that we are unclear about what has been learned from the training process, the knowledge, or how to do inference. To address this issue and encourage models to learn inference rather than knowledge from the training data, we propose to equip each question with supporting knowledge. Only if we can find supporting knowledge for a question will the question be selected to form the dataset. This section introduces the selected commonsense knowledge graphs and then introduces how we extract the corresponding commonsense knowledge for each question.

### 3.2.1 Commonsense KG Selection

Many commonsense knowledge graphs have been developed to enhance machines' commonsense reasoning abilities. Several representative ones are ConceptNet (Liu and Singh, 2004), ATOMIC (Sap et al., 2019), GLUCOSE (Mostafazadeh et al., 2020), and ASER (Zhang et al., 2020). Among these four, ConceptNet, ATOMIC, and GLUCOSE are constructed via crowd-sourcing while ASER is constructed automatically with information extraction techniques. Besides ATOMIC, which is used as one of the tasks, we use the other KBs as supporting knowledge resources.

### 3.2.2 Supporting Graph Extraction

Here we introduce how to extract the supporting knowledge from external commonsense knowledge bases. For each question, we need to obtain a sub-graph from supporting knowledge graphs such that it contains the relevant commonsense knowledge about the question. The sub-graph extraction process includes the following three steps: (1) Pre-processing: Convert each question into several key sentences; (2) Matching: Match the sentences into nodes in the KG; (3) Extraction: Retrieve the supporting sub-graph for the overall knowledge graph.

**Data Pre-processing**: For each question and the associated candidate answers, we first replace the question words (e.g., "What") with the two candidate answers such that it becomes two declarative sentences. For instance, if the question is "The fish ate the worm. It was hungry. Who is hungry?" and the candidates are "Fish" and "Worm," we will convert the question into the declarative sentence: "The fish is hungry" and "The worm is hungry." As a result, we will get three sentences for this question: "The fish ate the worm," "The

fish is hungry," and "The worm is hungry."

**KG Matching**: After getting the declarative sentences that contain the question and key answers, to extract the relevant knowledge, we map them to nodes in knowledge graphs. Considering that each sentence may have multiple words and it is often hard to find an exact match, we adopt an embedding-based matching technique. For each sentence and node in the KG, we treat them as a sentence and get the corresponding representations with SimCSE (Gao et al., 2021). For each input sentence, SimCSE encodes the sentence in a vector. A close distance between two vectors indicates that the two sentences are similar to each other. We use cosine similarity on the obtained representations to measure the similarity between two sentences.[1] Since there are 287 thousand nodes in GLUCOSE and 194 million nodes in ASER, it is computationally infeasible to compute the cosine similarity between sentences pair by pair. Thus for each extracted sentence, we first apply Faiss (Johnson et al., 2017), a large-scale similarity-based matching algorithm that first clusters all KG nodes in the vector space to increase the matching efficiency when finding the top $N$ nodes in the KG. After that, we sort the $N$ nodes based on the cosine similarity to find the top $K$ similar nodes. In our implementation, we set $N$ and $K$ to be 60 and 1. On average, it takes 25 seconds to retrieve relevant nodes for each question.

**Graph Extraction**: In the next step, we construct the sub-graph. We denote the extracted $m$ nodes as $n_1, n_2, ..., n_m$, and for each of them, we find $K$ similar nodes from KG. The resulting matched node sets are denoted as $\mathcal{N}_1, \mathcal{N}_2, ..., \mathcal{N}_m$. For any pair of eventualities $n \in \mathcal{N}_i$ and $n' \in \mathcal{N}_j$ ($i \neq j$), if there exist a path in the KG between $n$ and $n'$, we will keep that path. After merging all paths together, we will get the final sub-graph. On average, it takes less than two seconds to construct a graph for each question.

**Knowledge Quality Annotation**: We annotate whether the extracted knowledge is accurate and enough. For each question, we invite five annotators to provide the annotation. The average Inter-annotator agreement (Cohen's kappa statistic) is 0.83, which indicates the high-quality of our annotation. In the end, we apply a strict standard (at least four of five annotators need to vote for gold)

---

[1]We also tried other techniques such as string match, ROUGE (Lin, 2004), and BLEURT (Sellam et al., 2020), but found them to be either inaccurate or too slow for our scale.

| Task Name | # Instance by Knowledge Resource | | | # Total Instance | # Instance with Gold Knowledge |
|---|---|---|---|---|---|
| | ASER | ConceptNet | GLUCOSE | | |
| HardPCR | 2,030 | 202 | 2,143 | 4,375 | 670 |
| CommonsenseQA | 530 | 31 | 37 | 598 | 59 |
| COPA | 103 | 41 | 149 | 293 | 78 |
| ATOMIC | 5,655 | 212 | 3,466 | 9,333 | 2,200 |
| Total | 8,318 | 486 | 5,795 | 14,599 | 3,007 |

Table 1: **CIKQA** statistics. We report the number of instances supported by different knowledge resources and annotated high quality (i.e., Accurate and Enough) knowledge.

to select the gold knowledge. More annotation details could be found in Appendix Section A.

### 3.3 CIKQA Statistics

We report the dataset statistics in Table 1. In total, we collect 14,599 instances, and among which Hard PCR and ATOMIC provide the most questions because their original datasets are much larger than others. According to the annotation, 16.69% of the supporting knowledge graphs are gold knowledge. Based on our analysis, annotators hold a very strict standard for selecting the gold knowledge. For each task, we randomly split the dataset into training, development, and testing set with a standard 8:1:1 splitting. As a result, we get 11,678 training, 1,459 development, and 1,462 testing instances. More detailed statistics, and examples of **CIKQA** are presented in Appendix Section B and C, respectively.

## 4 The JointI Model

We introduce a transformer-based commonsense inference model as a strong baseline for **CIKQA**. Unlike previous works that acquire question and knowledge representations separately, we propose to combine them first and then acquire the representation jointly. As a result, we name our method as Joint Inference (**JointI**). As shown in Figure 2, given a question $Q$, two answers $A^1$ and $A^2$, and a supporting knowledge graph $\mathcal{G} = (h_1, r_1, t_1, w_1), ..., (h_n, r_n, t_n, w_n)$, where $h$, $r$, $t$, $w$ indicates the head, relation, tail, weight respectively, and $n$ is the number of edges, our goal is predict which answer is the correct one. Here, all questions, answers, heads and tails in the KG are list of tokens. **JointI** consists of two main components (i.e., knowledge sampling and joint inference). Details are as follows.

**Knowledge Sampling:** As current language models require the input to be in a sequence format rather than a graph, we first conduct a weighted

random walk over $\mathcal{G}$ to convert it into several knowledge paths $\mathcal{P}$ that are in the format of sequence. During our sampling, the weight of an edge determines the possibility of it being sampled. As a result, an edge with a larger weight is more likely to appear in the sampled path and has a bigger impact on the prediction. Another point worth mentioning is that, following previous work (Lv et al., 2020), we convert all the relations into natural language according to the relation template (e.g., "IsA" to "is a"). As shown in Figure 2, each $P \in \mathcal{P}$ can be viewed as a long sentence, where nodes in $\mathcal{G}$ are connected with connectives. An example is "I sleep because I am tired so I rest on a bench...".

**Joint Inference:** The key difference between **JointI** and previous works is that we jointly acquire the representation of the knowledge, question, and answer rather than acquiring them separately and then combining. Many previous works have demonstrated the superiority of such an approach on other NLP tasks (Huang et al.; Sakaguchi et al., 2020). Specifically, if we want to predict the plausibility score for $A$ given $Q$, for each knowledge path $P$, we first concatenate it with the question $Q$ and candidate answer $A$:

$$S = [P : Q : A], \quad (1)$$

where $[\cdot]$ indicates the concatenation. We follow previous works to insert a special token between $P$ and $Q$ and $Q$ and $A$. Once obtaining a concatenated input of $P$, $Q$ and $A$, we encode it using a transformer module $Trans$ and get a prediction score with a multi-layer perceptron module $MLP$ for a particular question and answer:

$$f(Q, A|P) = MLP(Trans(S)). \quad (2)$$

After that, we will get the final prediction with the average of all sampled paths:

$$F(Q, A) = \frac{\sum_{P \in \mathcal{P}} f(Q, A|P)}{|\mathcal{P}|}. \quad (3)$$

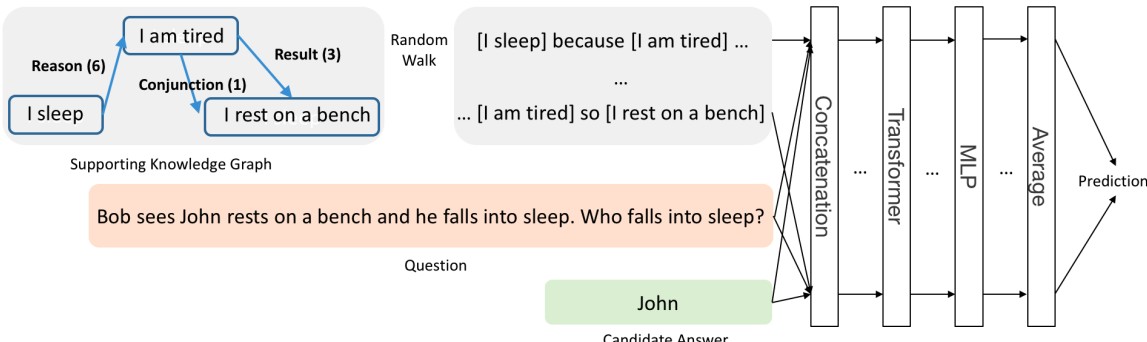

Figure 2: **JointI** demonstration. We first conduct weighted random walk over the supporting knowledge graph to sample several paths, and then concatenate these knowledge paths with the input question and answer together. In the end, we made the prediction with a transformer based classifier.

In the end, the candidate answer with a higher score will be predicted. Since the task is formulated as a binary classification problem, we adopt the cross-entropy loss and optimize the model with Adam (Kingma and Ba, 2015).

## 5 Experiments

In this section, we present the performance of current commonsense inference models on **CIKQA**. Besides **JointI**, we also show the performance of the following baseline methods:

**(1) Vanilla LM**: We use the language model (LM) based multiple-choice (MC) model as the basic baseline. For each candidate answer, we follow the standard finetuning procedure to concatenate it with the question and then feed it to a pre-trained language model. After getting the sentence representation, a linear layer is used to obtain a score and trained with a cross-entropy loss.

**(2) KagNet**: As one of the pioneering works that utilized structured knowledge for solving commonsense reasoning tasks, KagNet (Lin et al., 2019) first uses a graph convolution network to encode the knowledge graph and then apply an LSTM based hierarchical attention mechanism to encode the knowledge path that starts with the concepts corresponding to the question and end with concepts corresponding to the answer. At the same time, KagNet encodes the question and answers with pre-trained LMs. In the end, it concatenates all representations for the final prediction.

**(3) Graph Based Reasoning (GBR)**: Instead of only encoding paths starting with the question concepts and ending with answer concepts, the follow-up work GBR (Lv et al., 2020) proposes to conduct a depth-first algorithm over the knowledge graph to generate a sequence of paths as the supporting knowledge paths.

**(4) Multi-Head Knowledge Attention (MHKA)**: To further utilize the knowledge, MHKA (Paul and Frank, 2020) uses a transformer network to model the paths from the question concepts and answer concepts, then concatenates the knowledge and context representation for the final prediction.

We implement all experiments with Huggingface (Wolf et al., 2019). We select BERT-base (Devlin et al., 2019) as the base language model for all models. The batch size is set to be 16. All models are trained for 10,000 steps[2], and the best-performing checkpoints on the dev set are evaluated. For our model, we set both the number of random walk paths and walk length to be five. Considering that the auto-extracted knowledge could contain noise or miss certain knowledge, we add a "gold knowledge" setting, where only examples with the gold knowledge are used for training and testing, for all models as the upper bound of their model. All other hyper-parameters are the same as the base language model. All models are trained with GTX 2080 and the average running time is 12 hours.

### 5.1 Results

For each model, we train it with different numbers of training instances and report the average performance and standard deviation[3] of five trails in Figure 3, from which we can observe that with the help of knowledge, all inference models outperform the baseline model without knowledge, especially **JointI**. When the auto-extracted knowledge and gold knowledge are provided, **JointI** outperforms the baseline Vanilla LM model by 4.17

---

[2]All models converge at 10,000 steps.

[3]Due to the space limitation, we put the detailed experimental results in Appendix Section D.

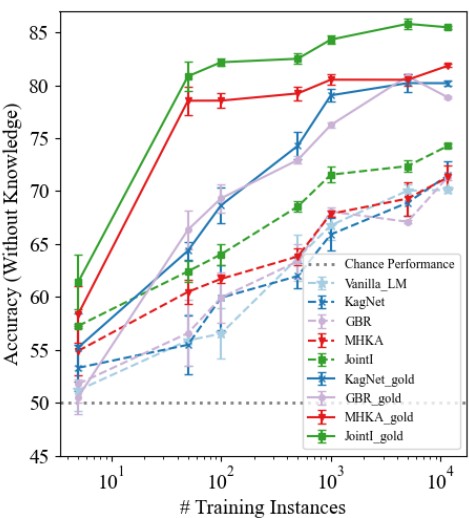

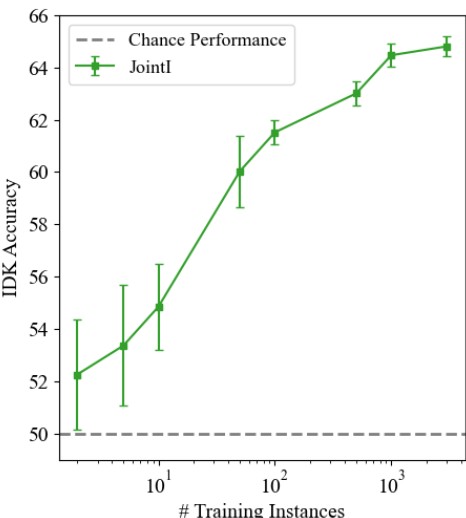

Figure 3: Learning curves of all evaluated models. Models with the "gold" suffix are evaluated on the gold subset of **CIKQA**, where only instances with gold knowledge are used for training and testing. We cannot directly compare them with other models, but they could serve as a good signal for the upper-bound of these models when we have a perfect commonsense knowledge base.

Figure 4: The learning curve of **JointI** on the gold knowledge identification task.

and 15.34, respectively. It supports our assumption that it is hard to learn all knowledge from the limited training data and external structured knowledge could help. Moreover, we also notice that when the knowledge is provided, **JointI** could learn to answer the questions with only a small number of examples. This suggests that if we only want to learn to do the inference over commonsense, we may only need a few training examples. Besides that, the comparison between auto-extracted knowledge and gold knowledge also shows that current commonsense knowledge base construction and retrieval methods are still not optimal and we may need to devote more effort to these two directions in the future. Last but not least, we can see that **JointI** outperforms other inference models among most settings, which shows that jointly encoding question and knowledge is not just more efficient but also a more effective strategy than acquiring them separately, and could serve as a stronger baseline for future works. Due to the simplicity and efficiency of **JointI**, we will conduct the rest analysis experiments with **JointI**.

## 5.2 Distinguishing the Gold Knowledge

Humans have the capability of saying "I do not know" when they find out that they cannot answer

a question with their knowledge. To investigate whether current deep models have a similar capability, we use **JointI** as an example to test whether these deep models can distinguish the gold knowledge. For each (question, answer, and knowledge) triplet, we train and test **JointI** with annotated knowledge quality label. To address the imbalanced distribution problem, we randomly select the same number of "Not Gold" examples as the "Gold" ones to make the dataset balanced. From the results in Figure 4, we can see that the performance of **JointI** could be improved slightly with the increase of training data. However, after seeing thousands of examples, it still can only achieve 0.65 accuracy on a binary classification problem. It shows that knowing when to say "I do not know" is still a challenging task for current deep models.

## 6 Generalization Ability

An important assumption and motivation behind **CIKQA** is that even though the commonsense could be enormous, the inference rules over commonsense knowledge should be limited. As a result, even though we could not learn all the commonsense from limited training data, we can learn how to conduct inference with several tasks and then generalize to others. In this section, we conduct experiments with both the "Without Knowledge" and "With Knowledge" models to show that with our unified formulation, we can gain such generalization ability across different tasks. To clearly show the effect of the supporting common-

| Training Task | Testing Task | | | |
|---|---|---|---|---|
| | Hard PCR | CommonsenseQA | COPA | ATOMIC |
| Hard PCR | - | 46.67/37.50 | 63.33/75.00 | 51.85/44.13 |
| CommonsenseQA | 49.32/50.00 | - | 50.00/62.50 | 60.39/56.34 |
| COPA | 52.51/45.95 | 56.67/62.50 | - | 53.01/49.77 |
| ATOMIC | 50.46/39.19 | 68.33/50.00 | 56.67/62.50 | - |

(a) Vanilla LM (Without Knowledge)

| Training Task | Testing Task | | | |
|---|---|---|---|---|
| | Hard PCR | CommonsenseQA | COPA | ATOMIC |
| Hard PCR | - | 51.67/52.30 | 56.67/53.24 | 55.78/53.32 |
| CommonsenseQA | 50.32/50.14 | - | 75.00/56.67 | 91.08/70.56 |
| COPA | 54.79/51.26 | 87.50/58.33 | - | 76.06/62.96 |
| ATOMIC | 51.35/50.76 | 93.75/76.67 | 87.50/73.33 | - |

(b) **JointI** (With Knowledge)

Table 2: Generalization ability demonstration. We report the performance on both the clean dataset (i.e., only questions with gold knowledge are selected for training and testing) and full dataset to show the generalization ability before and after the slash, respectively. Strong and moderate generalization settings are indicated with the green and orange background, respectively.

sense KB, we conduct experiments on two settings: (1) Gold Subset: We only train and test the model on questions, where the supporting graph is annotated as gold; (2) Full Set: We train and test the model with the whole dataset. We train the model with questions from a specific task and test it on all tasks. The results are presented in Table 2.

From the results, we can see that the knowledge can help models to generalize well among CommonsenseQA, COPA, and ATOMIC. The only exception is HardPCR. This is mainly because the inference needed for solving HardPCR is more complex than the other tasks, where we do not only need to find the relevant knowledge but also need to replace the target pronoun with the entity in the provided knowledge. How to train a model that can learn to conduct such complex reasoning is a problem worth exploring in the future.

In general, the observed generalization ability is encouraging because if we can learn a good model on **CIKQA**, based on the assumption that there exists limited types of inference, potentially we can solve any commonsense reasoning tasks as long as the needed inference type is covered by **CIKQA**. At the same time, we also notice that current models still cannot learn complex inference (i.e., compare multiple paths) with few examples, and we leave how to solve that problem as the future work.

## 7 Conclusion

In this paper, we present **CIKQA**, a unified commonsense inference benchmark. Specifically, we first convert several popular commonsense tasks into a unified QA format and equip each ques-

tion with a supporting commonsense knowledge graph. During the training on **CIKQA**, models do not need to worry about the commonsense knowledge and can thus focus on learning to do the inference. Experiments show that models can better learn how to do commonsense inference with a few examples and significantly outperform the baseline method that does not use structured knowledge in the data-scarce setting. More interestingly, with our unified formulation, models demonstrate the encouraging generalization ability across tasks. As both the format unification and supporting graph extraction are automatic, we can easily extend to other commonsense reasoning tasks in the future. All used code and data are submitted as part of the appendix.

## Acknowledgements

The authors of this paper were supported by the Office of the Director of National Intelligence (ODNI), Intelligence Advanced Research Projects Activity (IARPA), via IARPA Contract No. 2019-19051600006 under the BETTER Program, and by contract FA8750-19-2-1004 with the US Defense Advanced Research Projects Agency (DARPA). The views expressed are those of the authors and do not reflect the official policy or position of the Department of Defense or the U.S. Government. This paper was also supported by the NSFC Fund (U20B2053) from the NSFC of China, the RIF (R6020-19 and R6021-20) and the GRF (16211520) from RGC of Hong Kong, the MHKJFS (MHP/001/19) from ITC of Hong Kong with special thanks to HKMAAC and CUSBLT,

and the Jiangsu Province Science and Technology Collaboration Fund (BZ2021065).

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

## A  Annotation Details

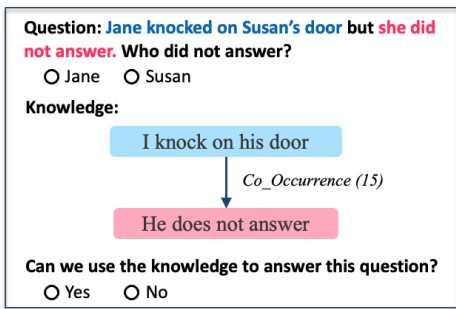

Figure 5: An example of the used survey.

The annotation goal is to determine whether the supporting graph can help answer the question or not. Thus, for each QA pair, we present the question, candidate answers, and the supporting sub-graph to annotators[4], and then ask them two questions: (1) What is the correct answer for this question; (2) Whether the provided commonsense knowledge contains all the essential commonsense for answering this question. The purpose of the first question is to assess the annotation quality. A survey example is shown in Figure 5. In beginning of each survey, we also provide detailed instructions and examples to help annotators understand our task. We employ annotators from Amazon Mechanical Turk to provide annotations. To improve the annotation quality, we require the annotators to be English native speaker and to have an overall acceptance rate above 90%. For each survey, we invite five annotators to provide the annotations and pay them $0.1. The average Inter-annotator agreement (Cohen's kappa statistic) for Q1 and Q2 are 0.87 and 0.83, respectively. The annotation results show that humans could provide consistent annotation about whether the knowledge could be used to answer the questions.

## B  Statistics

We report the number of questions that a supported graph can be find, the average size of supporting graph, and the number of helpful instances of **CIKQA** in Table 4. In total, we collect 14,599 instances with the average supported graph size of 2.75.

## C  Case Study

Demonstration of how we convert the original dataset into the unified format is presented in Table 3. For each task, we use a template to automatically convert it into the unified QA format. Besides that, we also present several questions along with knowledge in Figure 6. From the example we can see that, the reasoning over HardPCR is more challenging than other tasks. In the HardPCR example, two paths can be found relevant to question: (1) "I am drunk"→$Co\_Occurrence$→"I hit someone"; (2) "I am drunk"→$Co\_Occurrence$→"That is not fair"→$Co\_Occurrence$→"You kick me". For the correct inference, we need to know when there is a conflict, we should trust the one-hop inference more because the additional node in the two-hop path may introduce extra noise. As a comparison, for other tasks, the main inference we need is to find the relevant paths, which is relatively easy.

## D  Detailed Experimental Results

Detailed experimental results are presented in Table 5.

---

[4]All annotations follow the ethical guidelines.

| Task Name | Original Assertion | Transformed Question | Answer |
|---|---|---|---|
| HardPCR | The fish ate the worm. It was hungry. | The fish ate the worm. It was hungry. What was hungry? | (A) Fish; (B) Worm |
| CommonsenesQA | What is a place that someone can go buy a teddy bear? | What is a place that someone can go buy a teddy bear? | (A) Toy store; (B) Shelf |
| COPA | I drank from the water fountain. | I drank from the water fountain. What was the cause of this? | (A) I was thirsty.; (B) I felt nauseous. |
| ATOMIC | PersonX buys the bike. | Before PersonX buys the bike, what did PersonX want? | (A) To be social.; (B) To have transportation. |

Table 3: Demonstration of the original assertion, transformed questions, and answers. Correct and wrong answers are indicated with blue and red, respectively.

| Task Name | # Instances | Avg Sub-graph Size (# Edges) | # Helpful Instances |
|---|---|---|---|
| Hard PCR | 4,375 | 2.85 | 670 |
| CommonsenseQA | 598 | 3.19 | 59 |
| COPA | 293 | 3.03 | 78 |
| ATOMIC | 9,333 | 2.67 | 2200 |
| Total | 14,599 | 2.75 | 3,007 |

Table 4: Detailed **CIKQA** dataset statistics.

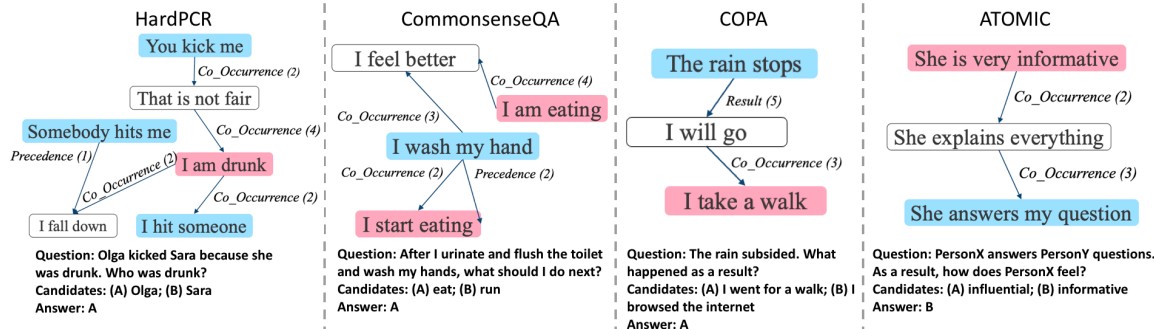

Figure 6: **CIKQA** Case Study. Mapped sentences for the question and answers are indicated with blue and pink. Other eventualities are white. Edge weights are in brackets. We only show the relevant part of the graph for the clear representation. All extracted eventualities are lemmatized, we recover them for the ease of understanding.

| Model | Number of Training Instances | | | | | | |
|---|---|---|---|---|---|---|---|
| | 5 | 10 | 100 | 500 | 1,000 | 5,000 | 11,678 |
| Chance Performance | 50.00 (0.00) | 50.00 (0.00) | 50.00 (0.00) | 50.00 (0.00) | 50.00 (0.00) | 50.00 (0.00) | 50.00 (0.00) |
| Vanilla LM | 51.16 (1.92) | 55.88 (2.41) | 56.52 (2.37) | 63.67 (2.19) | 66.76 (1.37) | 70.04 (0.58) | 70.11 (0.28) |
| KagNet (Lin et al., 2019) | 53.29 (2.16) | 55.47 (2.74) | 59.92 (3.05) | 61.97 (1.19) | 65.90 (1.54) | 68.90 (1.21) | 71.50 (1.29) |
| GBR (Lv et al., 2020) | 51.77 (1.75) | 56.57 (3.13) | 59.92 (2.34) | 63.36 (1.62) | 68.06 (0.35) | 67.10 (0.17) | 71.34 (0.31) |
| MHKA (Paul and Frank, 2020) | 54.89 (2.34) | 60.47 (1.13) | 61.70 (0.41) | 63.82 (0.78) | 67.85 (0.32) | 69.29 (1.58) | 71.30 (1.14) |
| **JointI**(Our Model) | **57.25** (0.21) | **62.41** (0.97) | **64.02** (0.99) | **68.54** (0.47) | **71.55** (0.75) | **72.36** (0.56) | **74.28** (0.21) |
| KagNet-gold | 55.21 (3.21) | 64.36 (0.83) | 68.65 (1.64) | 74.28 (1.31) | 79.05 (0.57) | 80.21 (0.84) | 80.20 (0.21) |
| GBR-gold | 50.53 (1.62) | 66.34 (1.82) | 69.31 (1.33) | 72.94 (0.35) | 76.24 (0.21) | 80.86 (0.21) | 78.85 (0.13) |
| MHKA-gold | 58.35 (2.67) | 78.54 (1.32) | 78.55 (0.72) | 79.23 (0.64) | 80.53 (0.50) | 80.52 (0.52) | 81.85 (0.15) |
| **JointI**-gold | **61.39** (2.56) | **80.85** (1.35) | **82.18** (0.33) | **82.51** (0.50) | **84.32** (0.42) | **85.81** (0.45) | **85.48** (0.17) |

Table 5: Demonstration of different models with different training instances. We report the average performance of five different random seeds and standard deviation (in brackets). "-gold" indicates that the models are trained and tested with instances with gold knowledge. We cannot directly compare them with the normal setting, but it could serve as the upper-bound for our learning paradigm. Best performing models under both settings are indicated with the **bold** font.