# OpenReview forum: "CIKQA: Learning Commonsense Inference with a Unified Knowledge-in-the-loop QA Paradigm"
_aclweb.org/ACL/2022/Workshop/CSRR — ACL 2022 Workshop CSRR_

### Official Review · Reviewer_MN7e · 2022-03-16

**Rating:** 6
**Confidence:** 3

**Review:**

The authors propose a reformulation of commonsense reasoning QA tasks
that attempts to separate knowledge (e.g., facts as specified by a KG)
from inference (i.e., reasoning over a given set of facts). Their
setup is to pair a small knowledge graph with each question that
contains the relevant knowledge to answer the question. They report
experimental results in this setting, showing that their model,
JointI, 1) effectively incorporates the knowledge graph information in
a fewer-shot setting (e.g., 100-1000 points); and 2) transfers between
tasks better than if the model didn't have explicit knowledge handed
to it.

I commend the authors for their attempt to solve a difficult problem:
indeed, the distinction between factual knowledge and inference over
that knowledge is rather illspecified in the commonsense domain.  The
proposed approach: converting and then augmenting existing QA datasets
with all the knowledge the might need gives a potentially nice
solution to this problem: i.e., by conditioning on "all of the
knowledge", the algorithms can focus entirely on inference; similarly,
by retrieving knowledge as a first step. I also think the results here
regarding generalization are quite interesting! Because we expect that
the inference required for commonsense reasoning tasks may be shared,
the transfer results suggest that, moreso than a model without
explicit knowledge provided, an inference-focused model may generalize
better.

My biggest concern is that I'm not entirely convinced that this setup,
as the authors claim, fully separates the knowledge versus inference
question. While the approach makes sense in theory (i.e., conditioning
on all needed knowledge), 1) there are still pieces of commonsense
knowledge required to, e.g., interpret the small KGs that are paired
with each question. To take the example in Figure 2: one simple case:
an algorithm must know that sleeping is a type of resting. And 2)
models could simply ignore the given knowledge graph in this setup,
e.g., if one was to use a pretrained language model that was already
imparted with both knowledge and inferential capacity. The authors do
use BERT-Small in some experiments and performance improves when the
graph is included in the input, but, I suspect that if more powerful
pretrained models were used the large performance gaps presented
between Table 2 (a) vs. Table 2 (b) might vanish.

Overall, the authors report some interesting results for their new
setup, which may have practical promise for few-shot learning with
small models. However, I do worry that CIKQA has limitations that need
to be addressed if larger models were to be applied to such a task.

---

### Official Review · Reviewer_FcT2 · 2022-03-23
**Commonsense benchmark paper - simple and well-presented**

**Rating:** 7
**Confidence:** 3

**Review:**

Overall the paper seems comparatively complete and solid to me. The authors propose the benchmark CIKQA with a clear task formulation and detailed steps on how to extract supporting knowledge, as well as a strong baseline that takes advantage of its format. Experiments are also well-executed to answer the authors' questions regarding leveraging provided knowledge, distinguishing gold knowledge, and model generalization ability on different tasks.

Here are my comments:
- It is a bit overclaimed to me that CIKQA can focus on learning to do the inference with the current task setting.
- The related work discussion seems a bit sparse to me.
- I am a bit concerned about the novelty as the unified format and injecting knowledge have all been discussed widely.
- From Table 1, there are only 3,007 instances with gold knowledge, but for experiment results in Figure 3, even models with the suffix `_gold` could still be trained with $10^4$ training instances. Hope the authors could address the issue there.

---

### Official Review · Reviewer_9YV6 · 2022-03-24
**a good benchmark paper overall**

**Rating:** 7
**Confidence:** 3

**Review:**

The paper proposes CIKQA, a commonsense benchmark, which unifies several commonsense task into QA format and associates them with relevant knowledge. Experiments shows that models can better learn inference and generalize across tasks with proposed formulation and usage of knowledge.

Strength:
1. the proposed benchmark can be a useful resource for the field.
2. the experiments and analysis are comprehensive and give interesting insights.
3. the writing is clear and easy to follow.

Weakness:
1. the coverage of proposed benchmark is limited, it doesn't include any physical or social commonsense tasks, like PIQA or SocialIQA.
2. the idea isn't entire novel - unified task formulation and knowledge injection have already been well-studied in QA domain.
3. missing related works:
[1] Liu, Jiachen et al. “Generated Knowledge Prompting for Commonsense Reasoning.” ArXiv abs/2110.08387 (2021): n. pag.
[2] Shwartz, Vered et al. “Unsupervised Commonsense Question Answering with Self-Talk.” ArXiv abs/2004.05483 (2020): n. pag.

---

### Decision · Program_Chairs · 2022-03-28

Accept